# Quality Traits of *Montanera* Iberian Dry-Cured *lomito* as Affected by Pre-Cure Freezing Practice

**DOI:** 10.3390/foods10071511

**Published:** 2021-06-30

**Authors:** David Tejerina, Lucía León, Susana García-Torres, Miriam Sánchez, Alberto Ortiz

**Affiliations:** Meat Quality Area, Center of Scientific and Technological Research of Extremadura (CICYTEX-La Orden), Junta de Extremadura, Ctra, A-V, Km372, 06187 Guadajira, Spain; david.tejerina@juntaex.es (D.T.); lucia.leon@juntaex.es (L.L.); susana.garciat@juntaex.es (S.G.-T.); miriam.sanchezo@juntaex.es (M.S.)

**Keywords:** *Montanera*, Seasonality, *Serratus ventralis*, Iberian *presa*, pre-frozen dry-cured *lomito*

## Abstract

The seasonality to which dry-cured products from Iberian breed pigs finished in *Montanera* (free-range rearing system with feed based exclusively on ad libitum consumption of natural resources; acorns and grass) are subjected could be overcome by pre-cure freezing. Three sets of *Montanera* Iberian *presas* (*Serratus ventralis* muscle) (*n* = 15) were established to assess the impact of frozen storage -0, or non-frozen, 3 and 6 months—previous to the technological process of curing—on the quality traits of the dry-cured product *Montanera* Iberian dry-cured *lomito*. Similar seasoning and curing processing conditions were applied to all sets. Lower productive performance due to higher weight loss during curing, and lower colour intensity were observed in pre-frozen dry-cured *lomitos*. The fatty acid profile was more saturated, and the oxidative status increased as a result of pre-cure freezing. On the matter of texture, all parameters were modified, highlighting the higher values of hardness and shear force of pre-frozen dry-cured *lomitos*. The time that raw material was frozen exerted a slight, thus helping manufacturers to better address the gap between industry and consumer demand with minimal effect on quality traits.

## 1. Introduction

Dry-cured products from Iberian breed pigs are valuable products, with those from pigs finished in *Montanera* system (free-range rearing in which feed is based exclusively on ad libitum consumption of acorns mainly from *Quercus ilex* and grass) attaining the highest organoleptic [1] and nutritional quality [2]. However, *Montanera* Iberian dry-cured products are subjected to seasonality due to the seasonal production of the natural resources of *dehesa* (rangelands with evergreen oaks and pastures found in the southwest of the Iberian Peninsula used for finishing *Montanera* Iberian pigs); acorns and grass (November to March). Thus, given that the highest consumption of Iberian dry-cured products occurs from November to December (owing to the Christmas season) there is a time gap between the commercial outlet of *Montanera* Iberian products and their consumers’ demand.

The freezing of the primal cuts prior to the technological process of curing would balance the production and demand of *Montanera* Iberian dry-cured products, and may help to overcome its implicit seasonality. Despite the technological and economic advantages it may have for the Iberian industry, the high commercial value and quality attained by *Montanera* Iberian dry-cured products [1] make it necessary to generate knowledge about the pre-cure freezing practice and its possible modifications in the quality traits and performance of the final product.

There are some specific designations of origin which do not contemplate such practice for products marketed under this designation. On the other hand, the European regulations for the freezing of animal products and the Spanish Iberian Quality Standard (IQS) do not mention this practice, which has led to a situation of uncertainty and lack of regulation and control of pre-cure freezing at industrial level. There is little scientific literature assessing the impact that pre-cure freezing practice has on quality characteristics of Iberian dry-cured products, which has been carried out on Iberian products from animals reared indoors and fed on commercial fodder (intensive system) [3,4,5,6] so their production could be distributed throughout the year, and therefore are less affected by the seasonality in consumer demand. Furthermore, most of the above-mentioned research studies were performed in Iberian dry-cured ham, for which the technological process of curing is lengthy, and therefore it is possible to adjust its commercial outlet to the needs of the market. However, the use of frozen/thawed counterparts to obtain dry-cured products is especially relevant for *Montanera* Iberian ones with a short technological process (60–70 days) which means that they would launch to the market in months with low commercial demand. In this line, only Ortiz, Tejerina, Contador, de Andrés, Petrón, Cáceres-Nevado, and García-Torres [7] addressed the quality traits of *Montanera* Iberian dry-cured loins as affected by the pre-cure freezing practice. 

The Iberian *presa* (*Serratus ventralis* muscle) is one of the most consumed primal cuts from Iberian pigs [8]. In addition to its consumption as fresh meat, Iberian *presa* can undergo a curing process, of a similar length to that followed by the loin, giving rise to Iberian dry-cured *lomito*, the commercial name given to the Iberian dry-cured *presa*, highly appreciated by consumers owing to its sensory characteristics. However, to the best of our knowledge, there is no literature dealing with Iberian dry-cured *lomito*.

Thus, the study of the pre-cure freezing practice in *Montanera* Iberian dry-cured *lomito* could contribute to the generation of knowledge of the factors that can condition the homogeneity of the Iberian dry-cured products and the standardisation of quality in general. Furthermore, it would specifically assess its effect on a product with short technological process, but with different physico-chemical characteristics to Iberian dry-cured loin [2] in an attempt to support the regulation and control of this practice at industrial level and by future legislative frameworks, as well as to shed light on the extent to which the quality of the *Montanera* Iberian dry-cured *lomito* could be affected.

Within this framework, the aim of the current research study was to assess the impact of pre-cure freezing practise (considering various timings of freezing storage; 3 and 6 months, for raw counterparts (*Montanera* Iberian *presa*)) on the quality traits of *Montanera* Iberian dry-cured *lomito*.

## 2. Materials and Methods

### 2.1. Meat Sampling

A total of 45 muscles *Serratus ventralis* (SV) were used in the present study. Muscles proceed from Iberian *Retinto* breed animals crossed with Duroc pigs (50:50) finished in *Montanera* system. 

Previous to finishing phase in *Montanera*, i.e., growing phase, animals were reared in open-air corrals and subjected to a feeding based on commercial feedstuffs. Thus, the average daily gain (ADG) during growing phase was 256 ± 28.3 (mean ± standard deviation; SD) g/day, whilst the average daily feed intake (ADFI) was 1.29 ± 0.08 kg/d. After that, animals started the finishing phase in *Montanera* system (open-air and free-range rearing system in which Iberian pigs are fed exclusively with fallen acorn and grass in the *dehesa agro-forestry* system) [9] with an average live weight of 102.9 ± 0.85 kg. The conditions animals were subjected to during *Montanera* phase were as follows.

*Montanera* was carried out in the *dehesa* of Valdesequera farm, (Badajoz, Spain). The length of *Montanera* was 67 days, comprised between November 2018 and January 2019. During this phase, animals were fed ad libitum on fallen acorns from *Quercus ilex* and grass. The animals had free access to water. The stock rate was 0.60 pigs per hectare, which was established with margins in order to guarantee that acorns would not run out before the fattening was completed [10]. At the beginning and at the end of the *Montanera* phase, the animals were weighted in order to calculate their ADG; 673.1 ± 34.9 g/d, attaining an average live weight of 148.4 ± 6.15 kg after finishing *Montanera*, with 14 months old. Subsequently, animals were transported to a local slaughterhouse (Mafrivisa, Castuera (6420), Spain). During transport animals were not supplied with feed or water [11]. During the period previous to slaughtering (less than 24 h), animals had access to water but not feed. Afterwards, animals were stunned by means of carbon dioxide and randomly slaughtered [12]. The consideration of ethical and welfare aspects by the Animal Care & Ethics Committee (ACEC) was not required for the development of the current study, because animals were subjected to standard husbandry practices during both the growing and the finishing stages, in compliance with the Council regulation Nº 2008/120 [13], the Royal Decree 1392/2012 [14] and the Royal Decree 1221/2009 [15].

After carcass quartering, which was carried out 4 h after slaughtering [16], SV muscles were removed from the left half of the carcass (*n* = 45). Afterwards, intermuscular fat and connective tissue were removed, and counterparts were individually packaged without vacuum in nylon/polyethylene bags (O_2_ permeability, 9.3 ml O_2_/m^2^/24 h at 0 °C), and cooled for 24 h (4 °C). Later, Iberian *presa* were allocated into three sets before curing: non-frozen (T0) (718.34 ± 88.16 g) (*n* = 15) and pre-frozen dry-cured *lomitos* for 3 (T3) (721.9 ± 69.95 g) (*n* = 15) (ii) and (iii) 6 (T6) (712.6 ± 116.37 g) months (*n* = 15).

### 2.2. Pre-Cure Freezing

The Iberian *presa* belonging to non-frozen (T0) set were immediately cured. Iberian *presa* from T3 and T6 sets were subjected to freezing and freezing storage during 3 and 6 months, respectively. Freezing was performed in a freezing chamber by means of cold air (−20 ± 2 °C) flowing at 20 km/h. Thus, the rate of freezing ranged from 1 to 5 cm/h. Frozen storage times and the freezing method was selected to be in line with the usual practices in the Iberian manufacturing industry. 

Thawing was conducted by placing the Iberian *presa* in a cold chamber at refrigerated temperature (4 ± 1 °C). Thereafter, the Iberian *presa* were subjected to curing process. The average weight of the Iberian *presa* after freezing storage for 3 or 6 months and subsequent thawing was 692.3 ± 67.81 g and 672.8 ± 113.81 g, respectively.

### 2.3. Technological Process of Curing

Seasoning, processing, and curing length were similar in the three loin sets and conducted according to regular practices of the Iberian manufacturing industry. Thus, all Iberian *presa* were seasoned with a mixture of salt (2.5%), paprika (0.6%) and 0.9% additives; E-250, E-252, and E-331. Subsequently, Iberian *presas* were stored at 4 °C for 48 h in darkness to allow the seasoning mixture to spread into the meat. Afterwards, Iberian *presa* were stuffed into 10 cm-diameter collagen casings and subjected to the technological process of curing. For the next 30 days they were kept at refrigerated temperature (4 °C) and high relative humidity (RH) (above 75%). During the rest of the time (to a total of 61 days), the temperature increased from 10 °C to 16 °C while the RH decreased to 60–65%. The average dry-cured *lomitos*’ weights according to pre-cure freezing sets were 439.1 ± 60.09, 395. 9 ± 47.02, and 373.4 ± 66.32 g for non-frozen (T0), pre-frozen dry-cured *lomitos* T3, and T6, respectively.

### 2.4. Methods

#### 2.4.1. Proximate Composition

The dry matter (DM) was conducted in accordance with standard method of the Association of Official Analytical Chemists [17].

Intramuscular fat (IMF) was extracted and quantified gravimetrically by means of chloroform/methanol (2:1, *v/v*) [18]. 

The chloride content (NaCl) was analysed by means of Volhard method [19].

#### 2.4.2. Weight Loss during Technological Process of Curing

The weight loss during curing was calculated as follows:% weight loss = ((W_0_ − W_f_)/W_0_) × 100,
being W_0_ is the weigh at the beginning and W_f_ the weigh after curing, respectively. 

#### 2.4.3. Instrumental Colour

Lightness (*L**), redness (*a**) and yellowness (*b**) coordinates were calculated using a Minolta CR-400 colorimeter (Minolta Camera, Osaka, Japan) with illuminant D65, a 0° standard observer and a 2.5 cm port/viewing area. The saturation index or chroma (*C**), defined as (C = (a^2^ + b^2^)^0.5^), and hue angle (H°) as arctangent (*b**/*a**) were calculated. The measurements were taken at five randomly selected sites on the steak of each sample (Iberian dry-cured *lomito*) and averaged.

#### 2.4.4. Antioxidant’s Content

α- and γ-tocopherol were extracted by means of a solution of KOH 11.5% in EtOH/H_2_O 55:45 [20]. Subsequently, identification and quantification of tocopherols was performed on an Agilent Technologies HPLC Series 1100 instrument (Agilent Technologies, Santa Clara, CA, USA), equipped with a Kromasil Silica column (5 µm particle size, 150 × 4.6 mm) (Symta, Madrid, Spain) and a Kromasil Silica Guard Column (10 µm) (Symta, Madrid, Spain), which conditions are fully described in Contador, Ortiz, Ramírez, García-Torres, and López-Parra [4].

#### 2.4.5. Fatty Acid Profile

Separation of main fatty acid methyl esters (FAMEs) was carried out from IMF previous extracted [18] by means of gas chromatograph (model 4890 Series II; Hewlett-Packard, Palo Alto, CA, USA) fitted with a Carbowax™ fused silica capillary column (30m×0.25 mm id; 0.25 µm film thickness; (Ohio Valley, Marietta, OH, USA). The oven, injector, detector, carrier gas conditions for individual FAME identification (throughout a standard mixture of 37 Component FAME Mix (Sigma–Aldrich, Supelco 37 Component FAME Mix- CRM47885, St. Louis, MO, USA)) are specified in Ortiz, García-Torres, González, De Pedro-Sanz, Gaspar and Tejerina [2]. Additionally, saturated, monounsaturated and polyunsaturated fatty acids groups; SFA, MUFA and PUFA, respectively were quantified.

#### 2.4.6. Lipid Oxidation

The oxidation of lipids was carried out by the 2-thiobarbituric acid (TBA) method [21].

#### 2.4.7. Protein Oxidation

For the purpose of protein oxidation, the carbonyl groups formed during incubation with 2,4-dinitrophenylhydrazine (DNPH) in 2N HCl was measured [22]. 

#### 2.4.8. Texture

Two texture analyses were performed; Texture Profile Analysis (TPA) and Warner-Bratzler test (WBSF) by means of a texturometer TA XT-2i Texture Analyser (Stable Micro Systems Ltd., Surrey, UK).

TPA was determined at 50% deformation [7,23] on uniform portions of 1 cm^3^ cubes of dry-cured *lomito*, which were axially compressed to 50% of it an original height with a 20 mm diameter (P/20) flat plunger using a 25-kN load cell applied at a crosshead speed of 2 mm/s through a 2-cycle sequence. Thus, hardness (N/cm^2^), springiness (cm), cohesiveness (dimensionless), gumminess (N cm s^2^), chewiness (N cm s^2^), and resilience (dimensionless) were determined from force–deformation curves [24].

For the WBSF test, samples—10 × 30 × 10 mm slices (width × length × thickness)—were cut with a Warner–Bratzler blade (HDP/BS) in a direction perpendicular to the muscle fibres. The maximum shear force (N/cm^2^) was measured to cut samples.

For both analyses, the mean value of each sample was obtained from eight replicates. 

#### 2.4.9. Statistical Analysis

The data were analysed using the statistical SPSS package SPSS.PC+ v. 20.0. The model included the pre-cure freezing time (T0, T3, and T6) as factor for the one-way ANOVA analysis of variance of variables related to the proximate composition, weight losses, instrumental colour, antioxidant and fatty acid profile, oxidative status and instrumental texture of the *Montanera* Iberian dry-cured *lomito*. Thus, the model used was as follows:Y_ij_ = µ + PCF_i_ + e_i(j)_
where Y_ijk_ is the variable considered; µ is the mean value; PCF_i_ is the effect of pre-cure freezing time (i =1: T0 or non-pre-cure freezing, i = 2: T3, i = 3: T6) and e_i(j)_ is the residual error.

The data were presented as mean ± standard deviation (SD). Tukey’s HSD test was applied to compare the mean values of each group. Statistical significance was set at *p* ≤ 0.05.

A principal component analysis (PCA) was performed by means of Unscrambler X software (CAMO^®^ Trondheim, Norway) to check the overall effect of the pre-cure freezing practice on *Montanera* Iberian dry-cured *lomito* samples distribution and to explore the multivariate relationships among variables.

## 3. Results

### 3.1. Proximate Composition, Weight Losses, and Colour Coordinates

With regard to proximate composition (Table 1), differences were observed for DM content (*p* ≤ 0.05) due to pre-cure freezing. Thus, DM was higher in *Montanera* Iberian pre-frozen dry-cured *lomitos* compared to those manufactured from non-frozen raw material. No effect was observed owing to freezing storage time of the *Montanera* Iberian *presas* (*p* > 0.05).

In refer to IMF content, values were comprised between 26.54 ± 2.23 and 27.83 ± 1.80 g/100 g of moisture-free dry-cured meat. The salt content value ranged from 2.74 ± 0.23 to 2.94 ± 0.50. Pre-cure freezing did not exert any significant differences in either of the two parameters.

The higher WL due to water losses during the technological process of curing was observed for *Montanera* Iberian pre-frozen dry-cured lomitos—irrespective of the previous freezing time of the raw material (T3 and T6)—with respect to *Montanera* Iberian dry-cured (T0) ones (*p* ≤ 0.05). 

Referring to the instrumental colour, redness (*a**) and yellowness (*b**) decreased due to the freezing of *Montanera* Iberian *presas* before curing irrespective of the freezing time (*p* ≤ 0.05). In contrast, no differences in lightness were observed due to pre-cure freezing (*p* > 0.05). Chroma and Hue differences might be explained by the modifications of the abovementioned coordinates, especially redness and yellowness. Specifically, Chroma was higher in dry-cured *lomitos* manufactured from non-frozen (T0) Iberian *presas*, whilst the pattern followed by Hue was not clear, with the highest value yielded by *Montanera* Iberian pre-frozen dry-cured *lomitos* (T3) (*p* ≤ 0.001).

### 3.2. Antioxidant Content, Fatty Acid Profile, and Oxidative Status

Results for the antioxidant composition as affected by pre-cure freezing practice are shown in Table 2. Thus, the α- and γ-tocopherol content was unaltered by pre-cure freezing practice (*p* > 0.05), with values ranging from 18.84 ± 0.84 to 20.14 ± 1.84 and from 2.48 ± 0.29 to 2.66 ± 0.30 for α- and γ-tocopherol, respectively.

The oleic acid (C18:1 n-9) was the most abundant fatty acid, yielding values from 51.57 ± 2.09 to 52.41 ± 1.66 followed by the palmitic acid (C16:0) 24.01 ± 1.00 to 25.51 ± 1.14 (Table 2). The pre-cure freezing influenced the SFA and PUFA groups, as well as some of the main individual fatty acids that comprise them; palmitic acid (C16:0) and linoleic (C18:2 n-6) and linolenic acids (C18:3 n-3), respectively (*p* ≤ 0.001). More specifically, *Montanera* Iberian pre-frozen dry-cured *lomitos* yielded lower values of both C18:2 n-6 and C18:3 n-3. Additionally, for the former, the time that raw material—Iberian *presas*—was frozen and stored influenced the ultimate value of this fatty acid (*p* ≤ 0.001). On the other hand, *Montanera* Iberian pre-frozen dry-cured *lomitos* revealed higher values of C16:0, with T6 ones reaching the highest value (*p* ≤ 0.001). The n-6/n-3 ratio was comprised between 8.80 ± 1.27 and 13.36 ± 2.37, revealing significant differences on account on the pre-cure freezing practice (*p* ≤ 0.001). Thus, an increase in the ratio was observed in *Montanera* Iberian pre-frozen dry-cured *lomitos*, irrespective of the time raw material was frozen before the technological curing process.

In the matter of oxidative status (Table 2), *Montanera* Iberian dry-cured *lomitos* manufactured from Iberian *presas* frozen/thawed yielded higher values of lipid and protein oxidation (*p* ≤ 0.001). 

### 3.3. Texture Analysis

Table 3 shows the effect of pre-cure freezing on TPA and WBSF tests on *Montanera* Iberian dry-cured *lomitos*.

The pre-cure freezing practice provided differences on all textural parameters evaluated in the TPA test (*p* ≤ 0.001). Thus, *Montanera* Iberian pre-frozen dry-cured *lomitos* yielded higher values for hardness and chewiness and lower values for springiness, cohesiveness, and resilience. Nevertheless, the time that raw material—*Montanera* Iberian presas—were stored frozen (3 or 6 months) did not derive in any differences for the above-mentioned parameters.

With respect to WBSF test, the pre-cure freezing practice resulted in higher values of shear force (*p* ≤ 0.001), regardless of freezing storage time of the *Montanera* Iberian *presas*.

### 3.4. Principal Component Analysis

The Figure 1 shows the results from the factorial analysis. The model clearly discriminated between *Montanera* Iberian pre-frozen dry-cured *lomitos* and dry-cured ones, but not between pre-frozen dry-cured *lomitos* according to freezing storage time of the raw material Iberian *presas*; T3 vs. T6. This splitting was mostly conducted on principal component (PC) 1, as it was responsible for more than 50% of the variance. Thus, samples from pre-frozen dry-cured *lomitos* placed in the left of the plot, mainly in the upper quadrant, whilst dry-cured *lomitos* samples from frozen/thawed Iberian *presas* tended to have positive scores in the PC 1 as well as in PC 2 axis, the latter explaining the 13% of the variance.

## 4. Discussion

The higher DM content observed in *Montanera* Iberian pre-frozen dry-cured *lomitos* with respect to dry-cured ones (T0) may be explained by their higher WL throughout the technological process of curing. These results were not unexpected, since proof has been shown of the higher amount of water released during curing process in *Montanera* Iberian pre-frozen dry-cured loins [7] and these from Iberian pigs indoors reared with fed based on commercial feedstuff [25]. The WL increase because of the use of frozen/thawed material may be associated to the damage caused in muscle fibres of the raw material owing to ice crystal formation (*Montanera* Iberian *presas*) [26]. The differences in WL due to pre-cure freezing practice suggest that the technological process of curing should be adjusted according to the raw material (non-frozen vs. frozen/thawed Iberian *presas*) to avoid the decrease in production yield.

The results obtained for DM, in the current research for *Montanera* Iberian dry-cured *lomito* were slight higher than those observed for *Montanera* Iberian dry-cured loins, which could be associated to the higher IMF content of the former [7,27]. To our knowledge, there is no scientific literature on the characterisation of Iberian dry-cured *lomito* neither on the evaluation of the influence that factors related to production or technological processing have on their quality characteristics to compare our results.

In refer to redness as affected by pre-cure freezing practice, the lower values observed for *Montanera* Iberian pre-frozen dry-cured *lomitos* may be attributed to the oxidation of red pigments. Thus, Iberian *presas* might have yielded a lower myoglobin due to freezing storage [28] so the respective pre-frozen dry-cured *lomitos* would expect to have lower a* than those elaborated from non-frozen raw material. Variations in redness coordinate is an aspect to consider because of the importance consumers assign to it in the case of cured products [29], so may affect purchasing decisions. With regard to the decrease in yellowness in *Montanera* Iberian pre-frozen dry-cured *lomitos*, the lack of correlation in the current study with the lipid oxidation [30], which increased with pre-cure freezing, might be due to various factors. Firstly, the differences in the lipid oxidation values identified in this study owing to pre-cure freezing might have not been enough as to allow to increase b* values. On the other hand, there are other factors with an impact on this coordinate such as moisture and salt content [31] and that could therefore condition its performance. The pattern followed by yellowness in the present study was in line with that obtained by Ortiz, Tejerina, Contador, de Andrés, Petrón, Cáceres-Nevado, and García-Torres [7] for *Montanera* Iberian pre-frozen dry-cured loin. 

In general terms, values of redness observed for *Montanera* dry-cured *lomitos* were higher than those reported for *Montanera* Iberian dry-cured loins, whilst yellowness was slight lower [27].

The lack of differences provided for α-tocopherol due to pre-cure freezing practice suggest the great stability of these antioxidants after pre-cure freezing, which is consistent with findings obtained by Ortiz, Tejerina, Contador, de Andrés, Petrón, Cáceres-Nevado, and García-Torres [7] in *Montanera* Iberian dry-cured loins. In this line, Martín-Mateos [23] previously proved the stability of α-tocopherol to freezing (through a freezing tunnel) and their subsequent freezing storage (−18 °C) in Iberian *presas*. Therefore, its stability after the curing process was expected.

On the other hand, in general terms tocopherols content values observed for *Montanera* Iberian dry-cured *lomitos* were higher than those identified in *Montanera* Iberian dry-cured loins [27]. This might be associated with differences in the values of both α- and γ-tocopherols of the raw material from which these dry-cured products originate; *longissimus thoracis et lumborum* and *Serratus ventralis* muscles, respectively [2]. More in detail, the high tocopherol content of *Serratus ventralis* could be due to its oxidative nature as opposed to the glycolytic one of the *longissimus thoracis et lumborum* [32].

The values observed for C16:0, C18:0 and C18:1 n-9 fatty acids in *Montanera* Iberian *lomitos* agreed in general terms with those previously reported for *Montanera* Iberian dry-cured loin, whilst C16:1 was lower and C18:2 n-6 and C18:3 n-3 higher [7,27]. 

The decrease in the unsaturation with pre-cure freezing was not unexpected. Thus, the decrease in PUFA group as well as the individual fatty acids that comprise it might be ascribed to lipolysis and lipid oxidation processes, since the oxidation susceptibility is dependent on the level of unsaturation of fatty acids [33,34]. Thus, given the more intense lipid oxidation observed in *Montanera* Iberian pre-frozen dry-cured *lomitos*, lower PUFA values and, in consequence, higher percentage of SFA would be expected. These results agree with those found by Ortiz, Tejerina, Contador, de Andrés, Petrón, Cáceres-Nevado, and García-Torres [7] for the case of *Montanera* Iberian dry-cured loin, although in the latter study the increase in SFA after pre-cure freezing did not reach a significant level. The differences observed for the n-6/n-3 ratio resulted from the variations of both C18:2 n-6 and C18:3 n-3, and in any case, the mean values attained were lower than those observed for the case of *Montanera* Iberian dry-cured loin [27].

On the matter of oxidative status, in general terms, higher rates of lipid oxidation were observed for *Montanera* Iberian dry-cured *lomitos* in comparison with those described for *Montanera* Iberian dry-cured loins [7]. This could be related to the muscle’s composition. Since the *Serratus ventralis* is an oxidative muscle, it may be more susceptible to oxidative processes as a result of factors related to technological processing than *Longissimus thoracis et lumborum* one [35], from which the loin is derived. These results might therefore suggest a different aptitude of Iberian dry-cured *lomito* vs. dry-cured loin to the pre-cure freezing practice. Additionally, the higher lipid oxidation values observed in *Montanera* Iberian pre-frozen dry-cured *lomitos* might be explained by the probably higher initial values of the raw material (Iberian *presas*) because of freezing storage [7,28]. Likewise, the protein oxidation values of the raw material together the higher products from lipid oxidation, as these can act as a substrate for the oxidation of proteins [36], might explain the higher protein oxidation of pre-frozen dry-cured *lomitos* with respect to dry-cured ones (T0). The higher extent of both lipid and protein oxidation of *Montanera* Iberian dry-cured *lomitos* as result of pre-cure freezing are consistent with results reported by Ortiz, Tejerina, Contador, de Andrés, Petrón, Cáceres-Nevado, and García-Torres [7] for *Montanera* Iberian dry-cured loins. Lorido, Ventanas, Akcan, and Estévez [4] also found higher protein oxidation in Iberian dry-cured loins manufactured from counterparts previously frozen for 5 months compared to those manufactured from fresh ones, but in the latter research study the oxidation lipid values remained unchanged because of the pre-cure freezing. On the other hand, the non-differences observed in oxidative state between T3 and T6, i.e., due to the time of frozen storage of the raw material, suggests that the oxidative state is altered by the freezing/thawing process itself, being of lesser significance the time the raw counterparts remain frozen.

Concerning instrumental texture properties, the higher hardness after pre-cure freezing (T3 and T6) might be related to the higher weight loss observed for *Montanera* Iberian pre-frozen dry-cured *lomitos*. Additionally, the higher protein oxidation observed for *Montanera* Iberian pre-frozen dry-cured *lomitos* might have enhanced the formation of cross-links [37] and protein aggregates, resulting in a strengthening of the product [38]. The pattern observed for hardness because of pre-cure freezing was in agreement with previous studies carried out on Iberian dry-cured loins from animals finished in *Montanera* [7] and indoors with fed based on commercial feedstuff [25]. Among the attributes related to texture analysis, hardness has been pointed out as the most relevant in terms of consumer acceptability of this type of products [39]. So, in view of the results obtained pre-cure freezing might influence consumer acceptability.

Mean values of hardness were in general lower than those reported for Iberian dry-cured loins [27]. Discrepancies between these two Iberian products might be due to various factors such as the differences in IMF contents between both muscles; *Serratus ventralis* and *Longissimus thoracis et lumborum* [2], the type and proportion of muscle fibres and collagen content [40].

The increase in shear force in the WBSF test with pre-cure freezing practice disagreed with previous scientific literature. Thus, Martín-Mateos [28] reported a decrease of this parameter on Iberian *presas* after the freezing storage, although this latter author did not study the texture properties after curing process. Ortiz, Tejerina, Contador, de Andrés, Petrón, Cáceres-Nevado, and García-Torres [7] found lower values in WBSF in *Montanera* Iberian pre-frozen dry-cured loins compared to those manufactured from non-frozen raw material. Discrepancies among studies might derive from the differences in the myofibrillar structure and physic-chemical characteristics between muscles from both dry-cured *lomitos* and loins are obtained; *Serratus ventralis* and *Longissimus thoracis et lumborum,* respectively [40]. 

The mean values of *Montanera* dry-cured *lomitos* observed for WBSF were in general similar to those described in *Montanera* dry-cured loins [7]. To our knowledge, few studies have addressed the influence of pre-cure freezing on Iberian dry-cured products [3,7,25]. However, there are no studies on the characterization of Iberian dry-cured *lomito*, nor how its textural parameters are affected by technological or productive related factors to compare our results.

The variations in the quality parameters above-mentioned as a result of pre-cure freezing allowed the separation of the samples in the new space defined by the principal component (PC) 1 and 2 according to the pre-cure freezing effect. More in detail, the negative scores of *Montanera* Iberian dry-cured *lomitos* samples manufactured from non-frozen Iberian *presas* might be due to the negative loadings of redness (*a**), polyunsaturated fatty acids such as C18:2 n-6 and C18:3 n-3 and some textural parameters such as springiness, cohesiveness, and resilience, which yielded higher values in these samples in comparison with pre-frozen dry-cured *lomitos.* In contrast, the positive loadings of lipid, protein oxidation, WL, DM, saturated fatty acids, and some textural parameters such as hardness and shear force could have led to the pre-frozen dry-cured *lomitos* samples to positive scores along the PC1, confirming the higher values of these parameters after pre-cure freezing. Nevertheless, when the pre-cure freezing time was considered, the separation was not clear. The slight differences exerted by the freezing storage time of raw material (Iberian *presas*) previous to technological process of curing on dry-cured *lomito* could explain the inability of the model to graphically discriminate between dry-cured *lomito* samples belonging to T3 and T6. Indeed, only hue coordinate, C16:0 and C18:2 n-6 were affected by pre-cure freezing time (3 vs. 6 months).

## 5. Conclusions

The findings of the current research study contributed to the generation of knowledge regarding the influence of the latest manufacturing-related factors on the quality of *Montanera* Iberian dry-cured products, providing insight into the Iberian meat industry in order to overcome the seasonality to which these products are subjected.

Our findings suggest that the use of frozen/thawed counterparts to obtain dry-cured *lomitos* may solve the gap between the industry and consumer demand, but could impact the productive performance of these, given the higher weight loss during technological process of curing and dry matter of the pre-frozen dry-cured *lomitos.* Therefore, pre-cure freezing should be considered by manufacturers to adjust the technological process conditions and length to the technological aptitude of the raw counterparts. Additionally, the pre-cure freezing may exert an effect on some quality traits such as instrumental colour, fatty acid profile, oxidative status, and textural properties.

Further studies should study its impact on sensory attributes to explore whether this practice may affect the consumer’s acceptability. The effect of pre-cure freezing practice on shelf life should be addressed.

## Figures and Tables

**Figure 1 foods-10-01511-f001:**
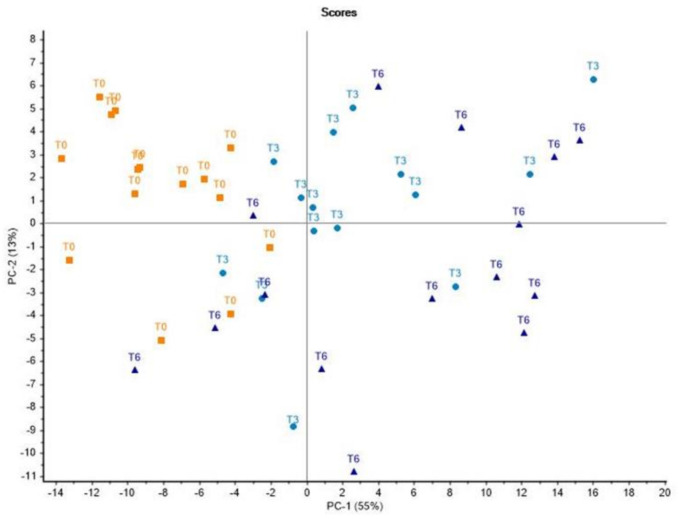
Projection of the samples onto the space defined by the first two principal components (PC1/PC2). *Montanera* Iberian dry-cured *lomito* samples were grouped according to pre-cure freezing effect (dry-cured *lomitos*; T0, and pre-frozen dry-cured *lomitos*; T3 and T6). (-) NaCl, (-) *a**, (-) *b**, (-) springiness, (-) cohesiveness, (-) resilience, (-) C16:1, (-) C18:1 n-9, (-) C18:2 n-6, (-) C18:2 n-3, DM, IMF, WL, *L**, α- and γ-tocopherol, lipid and protein oxidation, WBSF, hardness, chewiness, C16:0, C18:0 (PC1). (-) α- and (-) γ-tocopherol, (-) lipid and (-) protein oxidation, (-) WBSF, (-) hardness, (-) springiness, (-) cohesiveness, (-) chewiness, (-) resilience, (-) C16:1, (-) C18:1 n-9, DM, IMF, NaCL, WL, *L**, *a**, *b**, C16:0, C18:0, C18:2 n-6, C18:3 n-3 (PC2). T0: *Montanera* Iberian dry-cured *lomitos* manufactured from non-frozen Iberian *presas*, T3 and T6: *Montanera* Iberian pre-frozen dry-cured *lomitos,* manufactured from frozen during 3 and 6 months, respectively, and thawed Iberian *presas*.

**Table 1 foods-10-01511-t001:** Proximate composition, weight loss and instrumental colour of *Montanera* Iberian dry-cured *lomito* as affected by pre-cure freezing time.

	T0	T3	T6	Sig
Proximate composition (g/100 g)				
DM	58.52 ^b^ ± 2.08	60.53 ^a^ ±1.94	60.80 ^a^ ± 2.87	0.017
IMF ^1^	26.54 ± 2.23	27.67 ± 3.74	27.83 ± 1.80	0.362
NaCl	2.93 ± 0.29	2.74 ± 0.23	2.94 ± 0.50	0.187
Water losses (g water/100 g muscle)				
WL	40.62 ^b^ ± 1.83	42.85 ^a^ ± 2.91	44.19 ^a^ ± 2.36	0.001
Instrumental colour				
*L**	33.17 ± 1.73	33.88 ± 2.51	33.53 ± 2.65	0.697
*a**	15.95 ^a^ ± 0.56	14.42 ^b^ ± 1.38	15.07 ^b^ ± 0.31	0.000
*b**	5.39 ^a^ ± 0.47	5.08 ^b^ ± 0.46	4.97 ^b^ ± 0.41	0.010
Chroma	16.84 ^a^ ± 0.68	15.30 ^b^ ± 1.35	15.96 ^b^ ± 0.31	0.000
Hue	18.67 ^b^ ± 0.86	19.40 ^a^ ± 2.05	18.14 ^b^ ± 1.52	0.000

Values are expressed as mean ± standard deviation. Sig, significance. ^1^ Expressed in g/100 g moisture-free dry-cured meat. DM, dry matter; IMF, intramuscular fat; NaCl, salt content; WL, weight loss during technological process of curing; *L**, lightness, *a**, redness. *b**, yellowness. Values with the same letter (a, b) indicate homogeneous subsets for *p* = 0.05 according to Tukey’s HSD test.

**Table 2 foods-10-01511-t002:** Antioxidant content, fatty acid profile, and oxidative status of *Montanera* Iberian dry-cured *lomito* as affected by pre-cure freezing time.

	T0	T3	T6	Sig
Antioxidant composition (µg/g)				
α-Tocopherol ^1^	18.84 ± 0.84	18.52 ± 1.21	20.14 ± 1.84	0.093
γ-Tocopherol ^1^	2.53 ± 0.26	2.48 ± 0.29	2.66 ± 0.30	0.166
Fatty acid profile (g/100 g FAMEs)				
C16:0	24.01 ^c^ ± 1.00	24.89 ^b^ ± 0.92	25.51 ^a^ ± 1.14	0.000
C16:1	2.60 ± 0.29	2.69 ± 0.37	2.70 ± 0.41	0.636
C18:0	12.06 ± 0.72	12.57 ± 1.10	12.59 ± 0.67	0.082
C18:1 n-9	52.41 ± 1.66	51.65 ± 1.87	51.57 ± 2.09	0.124
C18:2 n-6	8.01 ^a^ ± 0.64	7.63 ^b^ ± 0.18	7.05 ^c^ ± 0.76	0.000
C18:3 n-3	0.91 ^a^ ± 0.13	0.56 ^b^ ± 0.15	0.57 ^b^ ± 0.10	0.000
SFA	36.07 ^b^ ± 1.57	37.46 ^a^ ± 1.85	38.10 ^a^ ± 1.58	0.000
MUFA	55.01 ± 1.61	54.34 ± 2.19	54.27 ± 2.28	0.247
PUFA	8.92 ^a^ ± 0.70	8.19 ^b^ ± 0.22	7.62 ^c^ ± 0.77	0.000
n-6/n-3	8.80 ^b^ ± 1.27	13.63 ^a^ ± 2.37	12.37 ^a^ ± 2.52	0.000
Oxidative status				
Lipid oxidation (µg MDA/g)	1.39 ^b^ ± 0.26	2.66 ^a^ ± 0.54	2.38 ^a^ ± 0.74	0.000
Protein oxidation (nmol carbonyls/mg protein)	1.59 ^b^ ± 0.74	2.04 ^a^ ± 0.32	2.19 ^a^ ± 0.36	0.000

Values are expressed as mean ± standard deviation. Sig, significance. ^1^ Expressed in g/100 g moisture-free dry-cured meat. FAMEs, fatty acid methyl esters; C16:0, palmitic acid; C16:1, palmitoleic acid; C18:0, stearic acid; C18:1 n-9, oleic acid; C18:2 n-6, linoleic acid, C18:3 n-3, linolenic acid, SFA: sum of saturated fatty acids (C16:0, C18:0); MUFA: sum of monounsaturated fatty acids (C16:1, C18:1 n-9); PUFA: sum of polyunsaturated fatty acids (C18:2 n-6, C18:3 n-3); n-6/n-3: PUFA n-6/PUFA n-3 ratio. MDA, malondialdehyde. Values with the same letter (a, b, c) indicate homogeneous subsets for *p* = 0.05 according to Tukey’s HSD test.

**Table 3 foods-10-01511-t003:** Textural properties of *Montanera* Iberian dry-cured *lomito* as affected by pre-cure freezing time.

	T0	T3	T6	Sig
Compression Test (TPA-50% compression)				
Hardness (N/cm^2^)	21.94 ^b^ ± 4.10	24.73 ^a^ ± 3.76	29.73 ^a^ ± 5.45	0.000
Springiness (cm)	0.69 ^a^ ± 0.02	0.63 ^b^ ± 0.04	0.61 ^b^ ± 0.09	0.000
Cohesiveness	0.64 ^a^ ± 0.03	0.60 ^b^ ± 0.05	0.57 ^b^ ± 0.06	0.000
Chewiness (N cm s^2^)	9.44 ^b^ ± 1.21	11.76 ^a^ ± 0.85	12.80 ^a^ ± 2.92	0.000
Resilience	0.25 ^a^ ± 0.02	0.21 ^b^ ± 0.03	0.22 ^b^ ± 0.02	0.000
Warner-Braztler shear force test (WBSF)				
WBSF (N/cm^2^)	23.24 ^b^ ± 2.84	32.20 ^a^ ± 8.31	35.45 ^a^ ± 12.79	0.001

Values are expressed as mean ± standard deviation. Sig, significance. TPA, Texture profile analysis. Values with the same letter (a, b) indicate homogeneous subsets for *p* = 0.05 according to Tukey’s HSD test.

## Data Availability

Data sharing not applicable.

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
