# Peer review of "Quality Traits of Montanera Iberian Dry-Cured lomito as Affected by Pre-Cure Freezing Practice"

_foods, 2021, doi:10.3390/foods10071511_

Round 1

Reviewer 1 Report

The manuscript is of general interest to the target public of Foods.

Abstract: The aim of the present should be clearly stated in the abstract. In line 21, please indicate whether the "slight effect" is positive or negative.

Both the analysed parameters and the description of the methodologies is adequate.

However, the experimental design is poor; the authors analysed 5 replicates of the same batch, that is, the technological process of curing was only performed once. This is in my opinion a major drawback, although I understand the technical and financial difficulties of repeating such an experiment twice.

Nevertheless, the authors assume this in the conclusions section, when they say that further studies are needed.

The generalised use of  hyphen "-" hyphen instead of commas to introduce a list of items within the text should be avoided. It reads with difficulty.

Author Response

Reviewer 1 (round 1)

 The manuscript is of general interest to the target public of Foods.

Abstract: The aim of the present should be clearly stated in the abstract. In line 21, please indicate whether the "slight effect" is positive or negative.

 Both the analysed parameters and the description of the methodologies is adequate.

However, the experimental design is poor; the authors analysed 5 replicates of the same batch, that is, the technological process of curing was only performed once. This is in my opinion a major drawback, although I understand the technical and financial difficulties of repeating such an experiment twice.

Nevertheless, the authors assume this in the conclusions section, when they say that further studies are needed.

The generalised use of hyphen "-" hyphen instead of commas to introduce a list of items within the text should be avoided. It reads with difficulty.

The authors really appreciate the reviewer's comment. The authors have slightly modified the abstract section, including a sentence regarding the possible benefits of the slight effect that the time of raw material is frozen has on quality traits of dry-cured lomitos. Additionally, authors have revised the manuscript to avoid the use of hyphen "-"

Reviewer 2 Report

The Article reports a study on the impact of pre-cure freezing practise -considering various timings of freezing storage (3 and 6 months) for raw counterparts (Montanera Iberian presa)- on the quality traits of Montanera Iberian dry-cured lomito. The paper fit with aims and scope of the Journal FOODS.

Lines 82-108: I recommend adding some References, especially to the method of breeding in what is different compared to other pigs for ham production.

Lines 101-102: "Afterwards, animals were stunned in compliance with the European Rules for the protection of animals during operations at the time of slaughter [10] by means of carbon dioxide and randomly slaughtered. " 

  • why carbon dioxide? It is necessary to define the method of production, whether it is in accordance with the procedure of traditional production, resp. whether this cannot affect the quality of the raw material.

Lines 110-114: "After carcass quartering, which was carried out 4 h after slaughtering according to Iberian standard commercial procedures."

  • why after 4 hours? Carcass quartering is common after the carcass is completely cooled (below 10 ° C). Has this been achieved? It is necessary to complete the citation/references to document the usual manufacturing/traditional process.

Lines 135-139: Thus, all Iberian presa were seasoned in a mixing bowl with a mixture of salt (2.5%), paprika (0.6%) and 0.9% additives (authorised preservatives and stabilisers) specifically prepared for this type of meat products and kept at 4°C for 48 h in darkness to allow the seasoning mixture to spread into the meat. 

  • What are the authorized preservatives and stabilizers? It is necessary to put a complete list in the Methodology or to refer to papers/standards in References.

Lines 168-174: Instrumental colour: The measurements were repeated at five randomly selected sites on each sample and averaged. Does this mean that these five randomly selected sites were the same at other measurement times?

I have no comments on the results presented here, they are clearly stated and presented. I agree with the discussion and conclusions of this study, although I think that its practical results will not be applied. However, the data obtained and presented in the article are an experiment that deserves to be published.
The article should be minor revision (corrections to minor methodological errors and text editing).

Author Response

Reviewer 2 (round 1)

  • The Article reports a study on the impact of pre-cure freezing practise -considering various timings of freezing storage (3 and 6 months) for raw counterparts (Montanera Iberian presa)- on the quality traits of Montanera Iberian dry-cured lomito. The paper fit with aims and scope of the Journal FOODS.
  • Lines 82-108: I recommend adding some References, especially to the method of breeding in what is different compared to other pigs for ham production.

The authors appreciate the reviewer's comment and interest for the method of breeding.

According to the reviewer's suggestions, the authors have added more information concerning the Montanera system; the open-air and free-range rearing system in the dehesa agro-forestry system (rangelands with evergreen oaks and pastures that are found in the southwest of the Iberian Peninsula used for extensive livestock farming with mixed-species grazing (beef cattle, sheep, and Iberian pigs)), in which Iberian pigs are finished.

Furthermore, authors have incorporated some references concerning the breeding of Iberian pigs in the final fattening phase in Montanera (References [9] and [10] of the manuscript).

  • Lines 101-102: "Afterwards, animals were stunned in compliance with the European Rules for the protection of animals during operations at the time of slaughter [10] by means of carbon dioxide and randomly slaughtered." why carbon dioxide? It is necessary to define the method of production, whether it is in accordance with the procedure of traditional production, resp. whether this cannot affect the quality of the raw material.

The authors appreciate the reviewer's comment and interest in the subject.

Carbon dioxide stunning is the common and most widespread method of stunning pigs before slaughter, not only for Iberian breed pigs [1,2], but also for other commercial breeds as demonstrates recent works [3]. The exact carbon dioxide concentration of the chamber and time that animals are maintained in that for proper stunning are private and specific data to each slaughterhouse and unfortunately the slaughterhouse is not willing to provide, although concentrations of 80-85% and exposure times of approximately 120-130 seconds are usually used. In any case, the authors do not consider this information essential to fulfil the objective of this manuscript as the initial material is fresh meat (Iberian presa) which was obtained from a batch of animals homogeneous in genetic background, age and slaughter weight at slaughter and under the same conditions of husbandry, slaughter and quartering operations.

Finally, the authors are not aware of whether this method of stunning could affect the quality of the meat derived. But in any case, as mentioned above, all pigs were slaughtered under the same conditions, so it is not an additional factor to be taken into account for the current research study.

  • Lines 110-114: "After carcass quartering, which was carried out 4 h after slaughtering according to Iberian standard commercial procedures." why after 4 hours? Carcass quartering is common after the carcass is completely cooled (below 10°). Has this been achieved? It is necessary to complete the citation/references to document the usual manufacturing/traditional process.

The authors appreciate the reviewer's comment. As it is stated in the manuscript, the carcass quartering is carried out in short-term after slaughtering (4 hours). This is known as “hot quartering”, and it is the way in which this operation is carried out in Iberian breed pigs (independently of the genetic background; purebred Iberian pigs vs. crossed with Duroc ones, the management system or the feeding of the animals). The hot quartering belongs to Iberian standard commercial procedures. Immediately after quartering, the products are stored in refrigeration chambers, where the temperature is 0-2ºC.

Specifically, with reference to the reviewer's comment, a reference has been included in material and methods section to support the practice of hot quartering.

In addition, the authors would like to emphasise the variety of published articles dealing with Iberian pigs in which it is stated that carcass quartering is carried out in short-term after slaughtering; [2,4–8]

  • Lines 135-139: Thus, all Iberian presa were seasoned in a mixing bowl with a mixture of salt (2.5%), paprika (0.6%) and 0.9% additives (authorised preservatives and stabilisers) specifically prepared for this type of meat products and kept at 4°C for 48 h in darkness to allow the seasoning mixture to spread into the meat. What are the authorized preservatives and stabilizers? It is necessary to put a complete list in the Methodology or to refer to papers/standards in References.

The authors appreciate the reviewer's comment. The authorized preservatives and stabilizers used for Iberian presas seasoning have been added to the manuscript. However, the specific formulation and therefore percentage of each one is exclusive and confidential of each manufacturer processing industry, which is not willing to share it.

  • Lines 168-174: Instrumental colour: The measurements were repeated at five randomly selected sites on each sample and averaged. Does this mean that these five randomly selected sites were the same at other measurement times?

The authors appreciate the reviewer's comment. For the instrumental colour measurement, five random measurements were taken with the colorimeter in a steak of each Iberian dry-cured lomito. These standards were followed for all samples for which colour was measured (samples from T0, T3 and T6), but obviously the sites were not the same, as they were random along the steak. Authors have tried to modified it in the manuscript.

  • I have no comments on the results presented here, they are clearly stated and presented. I agree with the discussion and conclusions of this study, although I think that its practical results will not be applied. However, the data obtained and presented in the article are an experiment that deserves to be published. The article should be minor revision (corrections to minor methodological errors and text editing).

The authors really appreciate the reviewer's comment. The authors have modified the manuscript according to the suggestions made by the reviewers.

References

  1. Ortiz, A.; González, E.; García-Torres, S.; Gaspar, P.; Tejerina, D. Do animal slaughter age and pre-cure freezing have a significant impact on the quality of Iberian dry-cured pork loin? Meat Sci. 2021, 179, doi:10.1016/j.meatsci.2021.108531.
  2. Ortiz, A.; Tejerina, D.; Contador, R.; de Andrés, A.I.; Petrón, M.J.; Cáceres-Nevado, J.M.; García-Torres, S. Quality Traits of Dry-Cured Loins from Iberian Pigs Reared in Montanera System as Affected by Pre-Freezing Cure. Foods 2020, 10, 48, doi:10.3390/foods10010048.
  3. Font-i-Furnols, M.; Luo, X.; Brun, A.; Lizardo, R.; Esteve-Garcia, E.; Soler, J.; Gispert, M. Computed tomography evaluation of gilt growth performance and carcass quality under feeding restrictions and compensatory growth effects on the sensory quality of pork. Livest. Sci. 2020, 237, doi:10.1016/j.livsci.2020.104023.
  4. Ortiz, A.; García-Torres, S.; González, E.; De Pedro-Sanz, E.J.; Gaspar, P.; Tejerina, D. Quality traits of fresh and dry-cured loin from Iberian x Duroc crossbred pig in the Montanera system according to slaughtering age. Meat Sci. 2020, 170, 108242, doi:10.1016/j.meatsci.2020.108242.
  5. Ortiz, A.; García-Torres, S.; González, E.; Contador, R.; Tejerina, D. Antioxidants and fatty acid deposition and instrumental colour as affected by subcutaneous backfat layers and slaughtering age of Iberian x Duroc crossed pigs under Montanera. Livest. Sci. 2020, 242, doi:10.1016/j.livsci.2020.104274.
  6. Ayuso, D.; González, A.; Hernández, F.; Peña, F.; Izquierdo, M. Effect of sex and final fattening on ultrasound and carcass traits in Iberian pigs. Meat Sci. 2014, 96, doi:10.1016/j.meatsci.2013.08.018.
  7. Ayuso, D.; González, A.; Peña, F.; Hernández-García, F.I.; Izquierdo, M. Effect of Fattening Period Length on Intramuscular and Subcutaneous Fatty Acid Profiles in Iberian Pigs Finished in the Montanera Sustainable System. Sustainability 2020, 12, doi:10.3390/su12197937.
  8. Fuentes, V.; Ventanas, S.; Ventanas, J.; Estévez, M. The genetic background affects composition, oxidative stability and quality traits of Iberian dry-cured hams: Purebred Iberian versus reciprocal Iberian×Duroc crossbred pigs. Meat Sci. 2014, 96 doi:10.1016/j.meatsci.2013.10.010.
